# Unraveling Hidden Representations: A Multi-Modal Layer Analysis for Better Synthetic Content Forensics

## Abstract

Generative models achieve remarkable results in multiple data domains, including images and texts, among other examples. Unfortunately, malicious users exploit synthetic media for spreading misinformation and disseminating deepfakes. Consequently, the need for robust and stable fake detectors is pressing, especially when new generative models appear everyday. While the majority of existing work train classifiers that discriminate between real and fake information, such tools typically generalize only within the same family of generators and data modalities, yielding poor results on other generative classes and data domains. Towards a "universal" classifier, we propose the use of large pre-trained multi-modal models for the detection of generative content. Effectively, we show that the latent code of these models naturally captures information discriminating real from fake. Building on this observation, we demonstrate that linear classifiers trained on these features can achieve state-of-the-art results across various modalities, while remaining computationally efficient, fast to train, and effective even in few-shot settings. Our work primarily focuses on fake detection in audio and images, achieving performance that surpasses or matches that of strong baseline methods.

## 1 Introduction

Recently, there has been an abundance of synthetic media emerging across various platforms and domains. Such media results from generative models based on deep neural networks, with novel techniques appearing at an unprecedented rate. Among these generative approaches, Generative Adversarial Networks (GANs) (Goodfellow et al., 2014) and diffusion-based tools (Sohl-Dickstein et al., 2015) produce high-resolution and highly-realistic results in multiple domains (Ho et al., 2022; Evans et al., 2024). While generative tools are often used for good purposes, they regrettably become tools for spreading misinformation and disseminating harmful deepfakes when exploited by malicious users. Thus, there is an increasing need for developing automatic approaches for identifying fake information that are robust across existing and future generative techniques. The overall goal of this paper is to advance existing knowledge and tools for automatic deepfake detection of multi-modal data, encompassing both visual and auditory modalities.

Early works for identifying manipulated images focused on highlighting visual inconsistencies such as irregular reflections, resampling, and compression artifacts (Popescu & Farid, 2005; O'brien & Farid, 2012; Agarwal & Farid, 2017). A similar line of works explored the frequency representation of images where inconsistencies of artificial media manifest as spectral fingerprints (Zhang et al., 2019; Frank et al., 2020). Recently, learning-based detection approaches emerged for classifying manipulated (Cozzolino et al., 2015; Rao & Ni, 2016; Wang et al., 2019) and generated (Marra et al., 2018; Rossler et al., 2019; Frank et al., 2020) images. Recent works of audio deepfake detection use carefully-chosen learning architectures, utilizing either sophisticated prepossessing (Li et al., 2021; Chen et al., 2020) or innovative deep network structures (Tak et al., 2021b; Ge et al., 2021). However, a few studies have found that while learned classifiers can be used for identifying synthetic media produced by a single class of generators (e.g., GANs) (Cozzolino et al., 2018; Zhang et al., 2019; Chen et al., 2020; Müller et al., 2022), they do not generalize to other classes (e.g., diffusion models). Thus, a fundamental challenge in deepfake detection is to develop an approach that has access to media from a single generator, yet it extends to various other generators (Nataraj et al., 2019; Wang et al., 2020a). Towards that end, our paper generalizes the work by Ojha et al. (2023) which suggested a deepfake detection

method that is effective across generative families of *visual content* by using certain features of CLIP-ViT models (Dosovitskiy et al., 2021; Radford et al., 2021).

To date, the detection of deepfakes of multi-modal information has received relatively limited attention in the literature, despite recent advancements in multi-modal learning that have demonstrated significant progress. Models like CLIP have developed shared latent spaces for images and text (Radford et al., 2021), while ImageBind and LanguageBind have extended this concept to multiple data modalities (Girdhar et al., 2023; Zhu et al., 2024). In the context of deepfake detection, recent studies have observed that the latent representations of CLIP-ViT can effectively distinguish between real and fake visual information (Ojha et al., 2023; Koutlis & Papadopoulos, 2024). Building on these findings, our work primarily addresses the research question: can latent representations of pre-trained large multi-modal models be effectively leveraged for deepfake detection across modalities, and if so, how? Existing works on images, arbitrarily select features from the last layer of the model (Ojha et al., 2023), or utilize features from all layers (Koutlis & Papadopoulos, 2024). The choice of last-layer features is natural from a machine learning perspective, as these features often exhibit linear separability (Goodfellow et al., 2016). However, CLIP-ViT was not explicitly trained to distinguish real from fake images but rather to align images with corresponding text. Consequently, the linear separability observed in CLIP-ViT's last-layer features likely reflects semantic relationships between language and vision (Gandelsman et al., 2023). Thus, the second research question we address is how to identify the optimal layers for multi-modal deepfake detection. In particular, we opt for a simple approach that requires minimal training, while effectively separating real and fake information.

To address the above questions, we propose leveraging the latent representations of large pre-trained multi-modal models for deepfake detection. Specifically, we argue that the most effective features arise from intermediate layers, rather than the initial or final layers. This approach is motivated by the observation that the initial layers of these models predominantly encode low-level details, akin to convolutional neural networks (Zeiler & Fergus, 2014), while the final layers capture high-level semantic information, particularly the last four layers in models like CLIP-ViT (Gandelsman et al., 2023). Consequently, we hypothesize that the "sweet spot" for optimal classification lies in the middle layers, where features balance low-level and high-level details. Our proposed method introduces a novel approach to deepfake detection by utilizing deep features extracted from the middle layers of large multi-modal models, rather than relying solely on features from the final layer or all layers. This straightforward yet effective strategy provides a unified solution across different modalities and generative approaches and it consistently outperforms or achieves similar results to recent approaches. The contributions of our work can be summarized as follows:

1. We extend the recent paradigm of deepfake detection in images, which leverages latent representations of large pre-trained models, to the multi-modal setting. Our work provides an in-depth analysis of such models, focusing on their separation properties across layers to understand their potential for detecting synthetic content.

2. Building on this analysis, we present a novel, unified, and universal multi-modal classifier that leverages intermediate representations for deepfake detection. Our classifier delivers a robust and efficient solution for the automatic identification of synthetic media, accommodating diverse families of generative models and spanning multiple modalities.

3. Our empirical results demonstrate the effectiveness of our approach over state-of-the-art tools on both image and audio modalities. Furthermore, we highlight several advanced detection capabilities, including clustering-based detection, source attribution and few-shot classification, showcasing the flexibility and versatility of our method.

## 2 Related Work

**Generative models.** Numerous works have been dedicated to advancing generative modeling across various data types including images, audio and video. Variational Autoencoders (VAEs) (Kingma & Welling, 2013) introduced foundational principles like approximate posterior estimation and the re-parametrization trick, extended further in Higgins et al. (2017); Van Den Oord et al. (2017); Gregor et al. (2019). Generative

Adversarial Networks (GANs) (Goodfellow et al., 2014) employ a zero-sum game in which a generating network tries to fool the discriminating network, yielding high-quality visual (Chen et al., 2016; Arjovsky et al., 2017; Brock et al., 2019; Karras et al., 2019) and auditory (Donahue et al., 2019; Kong et al., 2020) content. Diffusion models (Sohl-Dickstein et al., 2015) progressively add noise to an image and learn a denoising network, allowing to generate images matching in quality to GAN-based tools (Ho et al., 2020; Dhariwal & Nichol, 2021; Ramesh et al., 2022; Rombach et al., 2022). Additionally, diffusion has been adapted for generating varying-length audio (Evans et al., 2024) and video (Blattmann et al., 2023; Ho et al., 2022). Given that GAN- and diffusion-based methods are currently state-of-the-art (SOTA), we focus on works designed to identify their synthetic media.

**Deepfake detection.** Visual content generated by GAN techniques tend to leave noticeable traces in the frequency domain, whereas natural images do not exhibit similar patterns (Zhang et al., 2019; Wang et al., 2020a). Other works trained autoencoders to separate real and fake images (Cozzolino et al., 2018). Unfortunately, these approaches did not generalize well, even within the class of GANs (Nataraj et al., 2019). To this end, Wang et al. (2020a) developed an image classifier with blur and JPEG compression augmentations to improve detection accuracy. Still, generalizability was attained only to images sampled from the same family of generators. For audio deepfake detection, some approaches have successfully utilized wav2vec features (Baevski et al., 2020; Xie et al., 2021), while others have adopted end-to-end methods (Tak et al., 2021a). However, as highlighted by Müller et al. (2022), the persistent challenge of generalization remains a significant issue. Consequently, following works aimed for a "universal" detector: a deepfake detection method trained on one generator, but extends to multiple other (classes of) generators. For instance, subsequent works trained their classifier on images sampled from a diffusion model, while still detecting GAN images successfully (Ricker et al., 2022). Recently, the features extracted from the last layer and all layers of large vision-language models (CLIP-ViT) were shown to identify synthetic media across generative families (Ojha et al., 2023; Koutlis & Papadopoulos, 2024). However, to the best of our knowledge, no existing detector is capable of identifying multi-modal synthetic data generated by diverse families of generative models.

**Layer-wise analysis of deep networks.** Analyzing the inner mechanisms of modern neural networks is a longstanding problem (Zeiler & Fergus, 2014). Large vision models are often explained by generating heatmaps that emphasize areas of an image deemed most critical to the model's output (Binder et al., 2016; Selvaraju et al., 2017; Lundberg & Lee, 2017; Sundararajan et al., 2017; Voita et al., 2019; Chefer et al., 2021). Other approaches use the intermediate representations of deep models by inverting them to images (Mahendran & Vedaldi, 2015; Dosovitskiy & Brox, 2016), as well as interpreting neurons (Bau et al., 2019; 2020). A manifold learning viewpoint has been suggested for studying the properties of individual layers (Ansuini et al., 2019; Kaufman & Azencot, 2023). Recently, several techniques used text to interpret inner encodings of vision models (Materzyńska et al., 2022; Evan et al., 2022; Yüksekgönül et al., 2023). In particular, a technique for assigning attention heads of CLIP with text was proposed (Gandelsman et al., 2023), showing that the last four layers of CLIP are most dominant in image representation. While analysis of recent multi-modal models such as LanguageBind and ImageBind remains limited, we argue that their internal representations exhibit semantic properties akin to those of CLIP, given their similar training methodologies and architectural designs. This assumption drives our approach to exploit these representations for the unified and robust detection of fake information across multiple modalities.

**Concurrent deepfake tools.** Here, we also mention very recent approaches for images and audio. As for the image domain, Fernandez et al. (2023) embed an invisible watermark, allowing for future detection. Additionally, The DeepFakeFace dataset was introduced in Song et al. (2023), along with two evaluation methods. Fact checking was employed to verify whether generated images are consistent (Reiss et al., 2023). Another approach for image forensics was based on comparing rich and poor real and generated textures (Zhong et al., 2024). Recently, Cozzolino et al. (2024) showed that fewer samples can be used within CLIP models for a robust detection. As a final example for the image domain, we also mention an adversarial teacher-student discrepancy-aware framework (Zhu et al., 2023), achieving strong discriminative results. As for audio, Huang et al. (2023) utilizes the difference in high frequencies between real and generated audio for enhanced detection. Lastly we mention Guo et al. (2023), who utilized a large pre-trained audio model for

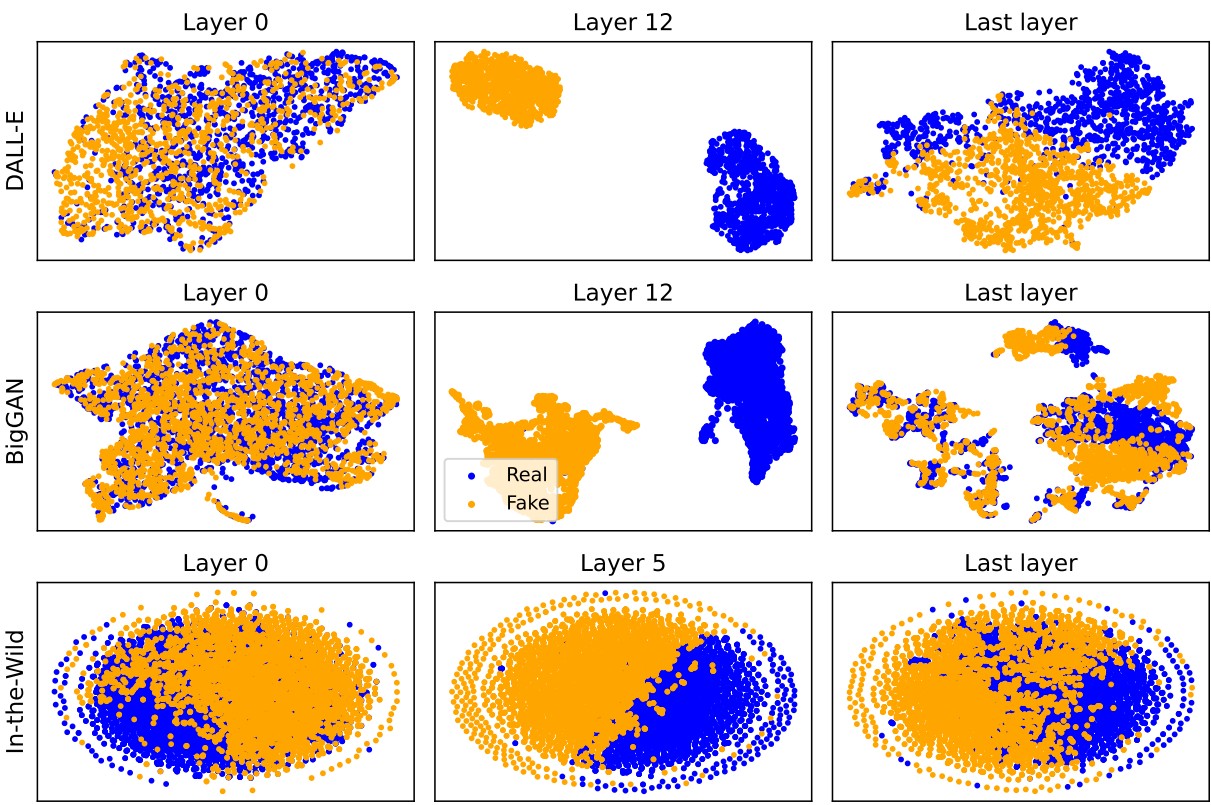

Figure 1: We plot two-dimensional t-SNE (pre-trained) embeddings of real (blue) and fake (orange) content related to the generative models BigGAN (top), DALL-E (middle), and In-the-Wild (bottom). Clearly, the intermediate features separate between real and synthetic media better than the first and last layers.

feature extraction, integrating these features with a specifically designed classifier and pooling method for effective deepfake audio detection.

## 3 Background

**Problem statement.** The problem we are set to solve in this paper can be defined as follows. Let the training set be composed of real content $\{c_{re}^i\}_M$ and fake content $\{c_{fa}^i\}_{\mathcal{M}}$ of modality $\mathcal{M}$, generated using a single generative model, where $i$ represents the sample index. We denote by $\mathcal{D}$ a deepfake detection model that takes a sample $c$ as an input, and returns a binary classification, specifying whether $c$ is natural or synthetic. To distinguish between generative models, we denote by $\{c_{fa}^{ij}\}$ as the dataset of fake content indexed by $i$ and generated by the method $j$. Then, our goal is that $\mathcal{D}$ attains the best measures on $\{c_{re}^i\}_{\mathcal{M}}$ as well as several $\{c_{fa}^{ij}\}_{\mathcal{M}}$, where some $j$ are of different classes of generative models, e.g., GAN and diffusion-based. Examples of evaluation measures include the accuracy and mean precision (Nataraj et al., 2019; Zhang et al., 2019) for images, and EER (Equal Error Rate) for audio (Lu et al., 2024).

**Multi-modal models.** The main idea behind the multi-modal models considered in this work is the learning of latent representations for various data modalities in a shared embedding space. One of the first approaches to achieve this was trained on a large collection of paired images and text, aligned using similarity matching, for example, by minimizing the cosine similarity (Radford et al., 2021). Recent advances have extended this concept, notably by incorporating multiple data modalities such as audio, depth, and thermal data, as demonstrated by LanguageBind and ImageBind (Girdhar et al., 2023; Zhu et al., 2024). These approaches are similarly trained with similarity matching, leveraging contrastive estimation, where

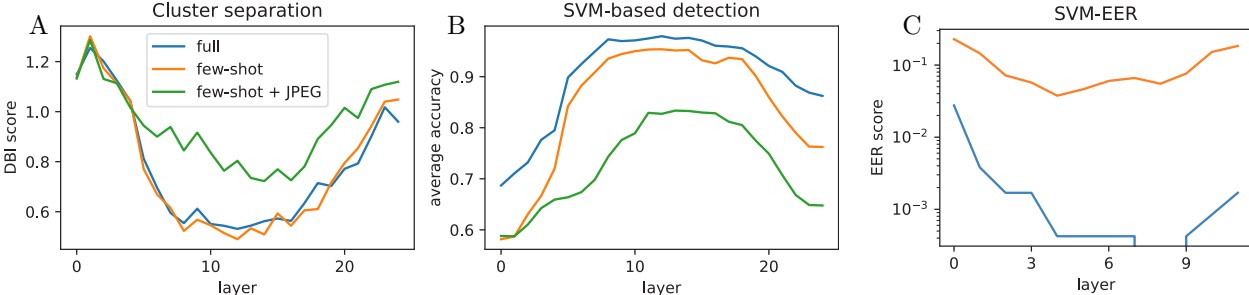

Figure 2: A) We compute the average Davies–Bouldin index score for latent representations of CLIP-ViT, showing that intermediate layers separate clusters better. B) We train SVM classifiers on latent features, and we demonstrate that first and last layers perform worse vs. middle layers. C) Similarly to B), however reporting the EER on In-The-Wild dataset using ImageBind latent features.

representations of paired samples are drawn closer in the embedding space while representations of unpaired samples are pushed apart. To formalize this, consider a paired sample, $c_{\mathcal{M}_1}^i$ and $c_{\mathcal{M}_2}^i$, drawn from modalities $\mathcal{M}_1$ and $\mathcal{M}_2$, respectively. The widely used InfoNCE loss (Oord et al., 2018) takes the form:

$$L_{\mathcal{M}_1,\mathcal{M}_2} = -\log\left(\frac{\exp\left(\mathbf{q}_i^\top \mathbf{k}_i/\tau\right)}{\exp\left(\mathbf{q}_i^\top \mathbf{k}_i/\tau\right) + \sum_{j\neq i}\exp\left(\mathbf{q}_i^\top \mathbf{k}_j/\tau\right)}\right), \tag{1}$$

where $\mathbf{q}_i$ and $\mathbf{k}_i$ are encoded representations of $c_{\mathcal{M}_1}^i$ and $c_{\mathcal{M}_2}^i$, respectively, and $\mathbf{k}_j$ represents the encoded representation of a dissimilar sample $c_{\mathcal{M}_2}^j$. The parameter $\tau$ is a temperature scaling factor. The summation in the denominator includes all unpaired samples, marked by the subscript $j \neq i$, which are dissimilar to $c_{\mathcal{M}_1}^i$. Trained in this way, multi-modal models enable various downstream tasks including cross-modal retrieval and zero-shot classification, demonstrating the flexibility of the embedding space across a broad set of inputs.

## 4 Analysis

In Ojha et al. (2023), the authors propose to utilize the features of the last layer of CLIP-ViT for deepfake detection. In contrast, Koutlis & Papadopoulos (2024) concatenated features from all layers. The following discussion highlights that either choice may be sub-optimal, toward addressing the question: What is the optimal multi-modal model layer for deepfake forensics? Similar to previous works, the underlying hypothesis in this paper is that synthetic media exhibit identifiable fingerprints. These artifacts may be noticeable to some extent in the frequency domain (Zhang et al., 2019; Frank et al., 2020). In this paper, unlike existing literature on this topic, we introduce another hypothesis, motivated by the roles of layers in CLIP-ViT. Specifically, we believe that the intermediate layers of CLIP-ViT extract improved digital fingerprints and frequency information for image forensics in comparison to the first and last layers. Additionally, we suggest that this behavior is not limited to CLIP-ViT, and it generalizes to other multi-modal frameworks when trained similarly, as proposed in Girdhar et al. (2023); Zhu et al. (2024). To verify our hypothesis, we consider a series of experiments, whose results guide our approach, as described in Sec. 5.

**2D t-SNE embeddings.** In this analysis, we extract features of real and synthetic samples from certain layers of the pre-trained CLIP-ViT model and ImageBind, and we view their distribution in a shared space. To this end, we extract features from the first, middle, and last layers. Then, we reduce their dimension using t-SNE (Van der Maaten & Hinton, 2008). We plot the resulting point clouds in a shared two-dimensional space, painting the real and fake points in blue and orange. Fig. 1 demonstrates the plots, where BigGAN (Brock et al., 2019) embeddings are shown in the top row, DALL-E (Ramesh et al., 2021) embeddings in the middle row, and In-the-Wild (Müller et al., 2022) audio embeddings in the bottom row. Importantly, Ojha et al. (2023) perform deepfake detection based on the features obtained from the last layer, where t-SNE struggles to separate real and fake information. In comparison, there is a clear separation between the two point clouds

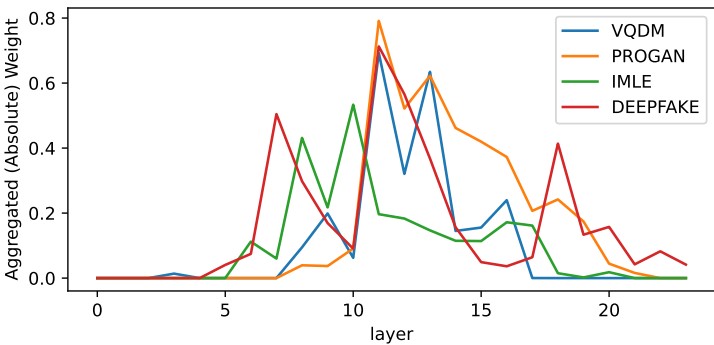

Figure 3: L1-regularized weights (absolute) values over layers of 4 classifiers, trained on VQDM, PROGAN, IMLE and DEEPFAKE with train splits from Ojha et al. (2023). The results, consistent across datasets and generative methods, demonstrate that the middle layers are most dominant for classification.

in the intermediate layer ('Layer 12/5'). We show in the appendix in Fig. 6 that a similar behavior appears for images from other GAN and diffusion generative models.

**Cluster separability.**   We extend the above analysis by estimating the separability of image clusters using machine learning tools. In this experiment (and in the next one), we consider three different scenarios of varying training sets: 1) we used 80% of the dataset (full); 2) we used only 15 examples (few-shot); and 3) we employ JPEG with a compression level of 50 (few-shot + JPEG). We computed k-means clustering over the features of every layer and every generative model in our benchmark. To measure separability, we estimate the Davies–Bouldin index (DBI) score (Davies & Bouldin, 1979) which is given by

$$\text{DBI} := \frac{1}{k} \sum_{i=1}^{k} \max_{j \neq i} \left( \frac{\sigma_i + \sigma_j}{d(c_i, c_j)} \right) \geq 0 \ , \tag{2}$$

where $k = 2$ is the number of clusters, $c_i$ is the centroid of cluster $i$, $\sigma_i$ denotes the mean distance of all points in cluster $i$ to $c_i$, and $d(\cdot, \cdot)$ is the Euclidean distance. A low DBI score means well-separated clusters. We illustrate in Fig. 2A the DBI scores per layer for the three scenarios mentioned above. Remarkably, the results coincide with our general hypothesis in that the first and last layers are challenging to discriminate, and intermediate features are well-separated. Finally, compressed synthetic images (green curve) are harder to separate, consistent with observations reported in Zhong et al. (2024).

**SVM-based detection.**   We also train SVM classifiers in the three configurations above (full, few-shot, and few-shot + JPEG), and we measure the average accuracy of real and fake media across all generative models in our image benchmark. Specifically, given a dataset $j$, we use the images $I_{\text{fa}}^{ij}$ as inputs to CLIP-ViT, and we compute their features $Z_{ij}^{l}$ for every image $i$ and layer $l$. The SVM classifier is trained on the features $Z_{ij}^{l}$ or their subset, depending on the particular configuration (full, few-shot, and few-shot + JPEG). We show the average accuracy vs. layer plots for the three scenarios in Fig. 2B. Similarly, we trained additional SVM classifiers on audio in full and few-shot setting, using ImageBind audio encoder as a feature extractor and report the EER (Equal Error Rate) per layer in Fig. 2C. Lower values are better. Our results show characteristic profiles, shared across all the scenarios. Particularly, the mean accuracy for image classification is low at the first and last layers, whereas performance is higher for intermediate layers. Surprisingly, utilizing only 15 images is sufficient to achieve $> 90\%$ mean accuracy (orange). Also, similar to the above experiment, compressed images yield the poorest overall detection results. The same trend repeats in the audio setting, whereas the EER is high in the first and last layers, and low at the intermediate layers in both settings.

**Layer-wise weight contribution.**   Finally, we conducted an analysis involving all layers of the model. Specifically, we concatenated the outputs from all layers and trained a classifier using this concatenated representation. To determine the importance of individual layers, we employed L1 regularization during training, which encourages sparsity in the learned weights. By analyzing the distribution of the absolute

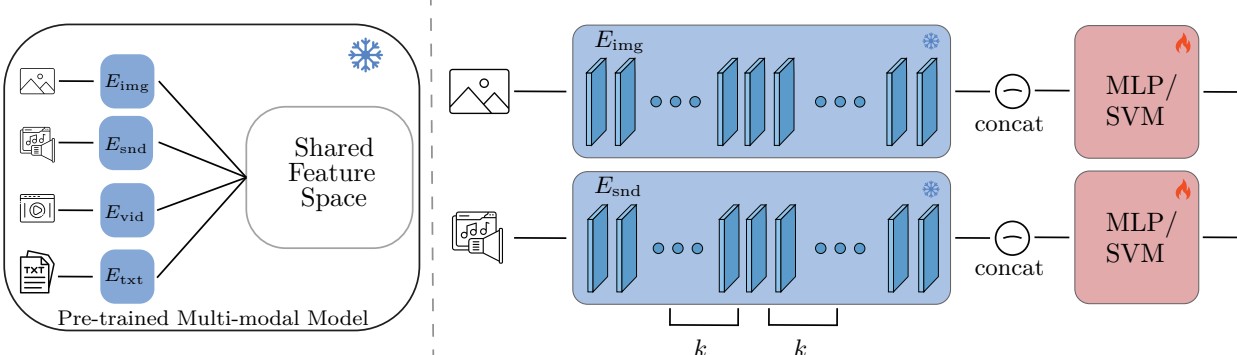

Figure 4: We leverage a pre-trained multi-modal model with one encoder per modality ($E_{\text{img}}, E_{\text{snd}}, \cdots$). Our deepfake detection approach extracts latent representations from the $2k + 1$ middle layers of the encoder, concatenates them, and inputs the combined features into a linear classifier.

values of these weights, we identified which layers contribute most significantly to the detection task. We summed the absolute value of each weight for each layer, shown in Fig 3. Aligning with our other analysis, the intermediate layers hold the highest weight values, while the first and the last layers are mostly sparse.

## 5 Method

We propose a framework for detecting synthetic content using a multi-modal model $M$, trained on modalities $m_1, m_2, \ldots, m_n$, each associated with a dedicated encoder $E_1, E_2, \ldots, E_n$. The model employs a contrastive loss $L$ to learn unified representations across modalities. To detect synthetic content in a specific modality $m_j$, we leverage the encoder $E_j$ corresponding to that modality. Let $l$ denote the total number of layers in $E_j$, and $h_{ji}$ represent the feature representation at the $i$-th layer. Our approach extracts latent representations $h_{ji}$ from $E_j$ as features and trains a lightweight classifier on them. During this process, the encoder $E_j$ is kept frozen, and only the classifier parameters are updated. To highlight the robustness of the proposed features, we employ simple classifiers, including a single-layer MLP and a linear SVM. The method is illustrated in Fig. 5, which outlines the architecture of the multi-modal model.

We hypothesize that the most discriminative features for this task are concentrated in the middle layers of $E_j$. To leverage this, we define a symmetric range of layers around the middle layer and select $k$ layers from this range as features, where $k$ is a hyperparameter. Specifically, if $k = 0$, only the middle layer representation $h_{\frac{l}{2}}$ is used. The value of $k$ is restricted to $0 \leq k \leq \frac{l}{2}$. Classifier training employs cross-entropy for the MLP and standard hinge loss for the linear SVM, underscoring the method's simplicity and effectiveness. We additionally introduce a simply method for choosing k at Appendix.A.

## 6 Experimental Setup

In this work, we limit our exploration to the detection of image and audio modalities. For images, we largely follow the experimental setup of prior works in the field (Wang et al., 2020a; Ojha et al., 2023). Specifically, the training set for our classifier is composed of real and synthetic images from ProGAN (Karras et al., 2018), while testing is conducted across multiple other datasets. The training set includes 20 distinct categories, each containing 18,000 synthetic images generated using ProGAN, alongside an equal number of real images sourced from the LSUN dataset (Yu et al., 2015). For audio, we use a diverse collection of generated audio samples produced by various methods, including In-the-Wild and ASVSpoof2019 (Müller et al., 2022; Wang et al., 2020b). As for pre-trained model, we consider ImageBind audio encoder. Below, we outline the baselines used for comparison and the evaluation metrics. It is worth noting that training of all classifiers across all modalities is performed for only one epoch.

**Generative models.** Many new generative models have been introduced over the past few years. Similar to existing image-based deepfake detection works, we focus our evaluation on GAN- and diffusion-based techniques. Specifically, we consider all the models used in Wang et al. (2020a): ProGAN (Karras et al., 2018), StyleGAN (Karras et al., 2019), BigGAN (Brock et al., 2019), CycleGAN (Zhu et al., 2017), StarGAN (Choi et al., 2018), GauGAN (Park et al., 2019), CRN (Chen & Koltun, 2017), IMLE (Li et al., 2019), SAN (Dai et al., 2019), SITD (Chen et al., 2018), and DeepFakes (Rossler et al., 2019). Additionally, we also test with respect to the techniques shown in Ojha et al. (2023): Guided (Dhariwal & Nichol, 2021), LDM (Rombach et al., 2022), Glide (Nichol et al., 2022), and DALL-E (Ramesh et al., 2021). We incorporate variants of LDM and Glide, as was introduced in Ojha et al. (2023). For LDM, three variants are compared: 200 steps, 200 steps and classifier-free guidance, and 100 steps; for Glide, different number of steps introduced in the two separate stages of noise refinement, creating three variants: 100-27, 50-27, and 100-10. Finally, we include two additional models shown in Zhong et al. (2024): StyleGAN2 (Karras et al., 2020), and WhichFaceIsReal (WFIR) (whi, 2019). Similarly to work on visual content, audio generation has evolved rapidly in recent years, and various generation methods have emerged. To validate robustness, we consider ASVSpoof2019 (Wang et al., 2020b); particularly its Logical Access (LA) part, which contains fake human speech generated with 19 different TTS models, and In-the-Wild (Müller et al., 2022) which extends ASVSpoof2019 with 'real-world' synthetic audio, containing fake audio samples from 58 politicians and celebrities.

**Deepfake detection baselines.** In our evaluation on images, we compare our results with respect to all the methods shown in Ojha et al. (2023), and further, we also consider some of the most recent state-of-the-art approaches for identification of synthetic content. CNNDet (Wang et al., 2020a) constructs a standard image classifier, trained with careful pre- and post-processing procedures and data augmentation. We use two variants of CNNDet with Blur and JPEG compression of levels 0.1 and 0.5, denoted as 'CNNDet1' and 'CNNDet2', respectively. The patch classifier (Chai et al., 2020) utilizes limited receptive fields, and we consider two different backbones for this architecture: ResNet50 ('PatchDet1') and Xception ('PatchDet2'). In Nataraj et al. (2019), CNN classifiers and co-occurrence matrices ('CO') are used. A GAN simulator was designed to produce the artifacts of synthetic media in Zhang et al. (2019) ('FreqSpec'). The Universal Fake Detector ('UFD') method by Ojha et al. (2023) is mostly related to ours as they extract the features of the last layer of a pre-trained CLIP-ViT model, and train a simple classifier using those features. RINE (Koutlis & Papadopoulos, 2024) leverages all CLIP-ViT layers, with an additional contrastive loss and a learnable layer-importance module. Finally, FreqNet (Tan et al., 2024) focuses on high-frequency information across spatial and channel dimensions, resulting in a frequency domain learning approach.

Our evaluation on audio data is based on comparing with the same models as in Müller et al. (2022). LCNN (Wu et al., 2020) learns a transformer with a convolutional neural network (CNN) using the characteristics of only genuine speech. RawNet2 (Tak et al., 2021a) is an end-to-end model. It employs Sinc-Layers (Ravanelli & Bengio, 2018), which correspond to rectangular band-pass filters, to extract information directly from raw waveforms. RawPC (Ge et al., 2021) is also based on Sinc-Layers to operate directly on raw wavforms, however, the model architecture is found via a novel differentiable architecture search. Finally, RawGAT-ST (Tak et al., 2021b) is a spectro-temporal graph attention network (GAT). It introduces spectral and temporal sub-graphs and a graph pooling strategy to discriminate between real and fake audio. We omit from our comparison works with no available code or weights, and works that are concurrent to ours.

**Evaluation metrics.** Similar to many existing works on this topic, we use two quantitative metrics to compare the performance of our approach with respect to other methods. The first metric, *accuracy* (ACC) (Wang et al., 2020a; Ojha et al., 2023), measures the ratio between the true classifications vs. the whole data. The second metric, the *mean precision* (mAP) (Lin et al., 2014; Zhou et al., 2018; Huh et al., 2018; Wang et al., 2019) measures the average of real and fake precision measures. The precision is calculated as the ratio between the true classifications of e.g., fake data vs. all data that was classified as fake. For audio-based detection, we show the EER (Equal Error Rate), similar to other works. EER quantifies the trade-off between the false acceptance rate and false rejection rate. It is the point on the receiver operating characteristic (ROC) curve where the two error rates are equal. A lower EER indicates better system performance.

Table 1: Deepfake detection on GAN-based approaches, listing measures in a ACC / mAP format.

| Method | ProGAN | CycleGAN | BigGAN | GauGAN | StarGAN | StyleGAN | StyleGAN2 | WFIR |
|---|---|---|---|---|---|---|---|---|
| CNNDet1 Wang et al. (2020a) | 100 / 100 | 85.2 / 93.5 | 70.2 / 84.5 | 78.9 / 89.5 | 91.7 / 98.2 | 87.1 / 99.6 | 84.4 / 99.1 | 83.6 / 93.2 |
| CNNDet2 Wang et al. (2020a) | 100 / 100 | 80.8 / 96.8 | 60.0 / 88.2 | 79.3 / 98.1 | 80.9 / 95.4 | 69.2 / 98.3 | 68.4 / 98.0 | 63.9 / 88.8 |
| PatchDet1 Chai et al. (2020) | 94.4 / 98.9 | 67.4 / 72.0 | 64.6 / 68.8 | 57.2 / 55.9 | 80.3 / 92.1 | 82.3 / 92.3 | - | - |
| PatchDet2 Chai et al. (2020) | 75.0 / 80.9 | 68.9 / 72.8 | 68.5 / 71.7 | 64.2 / 66.0 | 63.9 / 69.2 | 79.2 / 75.8 | - | - |
| CO Nataraj et al. (2019) | 97.7 / 99.7 | 63.2 / 81.0 | 53.8 / 50.6 | 51.1 / 53.1 | 54.7 / 68.0 | **92.5** / 98.6 | - | - |
| FreqSpec Zhang et al. (2019) | 49.9 / 55.4 | **99.9** / 100 | 50.5 / 75.1 | 50.3 / 66.1 | 99.7 / 100 | 49.9 / 55.1 | - | - |
| FreqNet Tan et al. (2024) | 99.6 / 100 | 95.8 / 99.6 | 90.5 / 96.0 | 93.4 / 98.6 | 85.7 / 99.8 | 90.2 / 99.7 | 88.0 / 99.5 | 51.1 / 49.2 |
| UFD Ojha et al. (2023) | 100 / 100 | 98.2 / 99.8 | 95.5 / 99.2 | 99.4 / 99.9 | 94.5 / 98.4 | 85.7 / 98.0 | 76.3 / 98.7 | 85.3 / 96.6 |
| RINE Koutlis & Papadopoulos (2024) | 99.8 / 100 | 99.5 / 100 | **99.1** / 99.9 | 99.5 / 100 | 97.6 / 99.9 | 90.2 / 99.2 | **89.8** / 99.5 | **98.3** / **99.9** |
| Ours (SVM$_9$) | **100** / **100** | 99.5 / **100** | 99.0 / **100** | **99.6** / **99.9** | **100** / **100** | 86.6 / **99.9** | 81 / **99.7** | 96.8 / 99.8 |
| Ours (MLP$_9$) | 100 / 100 | 99.5 / 100 | 98.8 / 99.9 | 99.4 / 100 | 100 / 100 | 82.6 / 100 | 73.3 / 99.1 | 96 / 99.8 |
| Ours (SVM$_0$) | 100 / 100 | 97.0 / 99.9 | 88.5 / 99.4 | 80.5 / 99.1 | 100 / 100 | 98 / 100 | 93 / 100 | 68 / 98.0 |
| Ours (MLP$_0$) | 100 / 100 | 95.0 / 100 | 94.9 / 99.9 | 93.7 / 99.7 | 88 / 100 | 98.7 / 100 | 94.1 / 100 | 92.4 / 99.0 |
| Ours (SVM$_{max}$) | 100 / 100 | 99.5 / 100 | 98.5 / 100 | 100 / 100 | 100 / 100 | 89 / 100 | 80.5 / 100 | 98 / 99.8 |
| Ours (MLP$_{max}$) | 100 / 100 | 99.5 / 100 | 99.5 / 100 | 100 / 100 | 100 / 100 | 90.5 / 100 | 83.5 / 100 | 98 / 99.9 |

Table 2: Deepfake detection on diffusion and autoregressive methods, where the top part details accuracy measures and the bottom part lists mean precision estimates.

| Method | Deep fakes | Low level vision | | Perceptual loss | | Guided | LDM | | | Glide | | | DALL-E |
|---|---|---|---|---|---|---|---|---|---|---|---|---|---|
| | | SITD | SAN | CRN | IMLE | | 200 steps | 200 w/ CFG | 100 steps | 100 27 | 50 27 | 100 10 | |
| CNNDet1 | 53.5 | 66.7 | 48.7 | 86.3 | 86.3 | 60.1 | 54.0 | 54.9 | 54.1 | 60.8 | 63.8 | 65.6 | 55.6 |
| CNNDet2 | 51.1 | 56.9 | 47.7 | 87.6 | 94.1 | 51.9 | 51.3 | 51.9 | 51.3 | 54.4 | 56.0 | 54.4 | 52.3 |
| PatchDet1 | 55.3 | 65.0 | 51.22 | 74.3 | 55.1 | 65.1 | 79.1 | 76.2 | 79.4 | 67.0 | 68.5 | 68.0 | 69.4 |
| PatchDet2 | 75.5 | 75.1 | **75.3** | 72.3 | 55.3 | 67.4 | 76.5 | 76.1 | 75.8 | 74.8 | 73.3 | 68.5 | 67.9 |
| CO | 57.1 | 63.1 | 55.9 | 65.7 | 65.8 | 60.5 | 70.7 | 70.6 | 71.0 | 70.3 | 69.6 | 69.9 | 67.6 |
| FreqSpec | 50.1 | 50.0 | 48.0 | 50.6 | 50.1 | 50.9 | 50.4 | 50.4 | 50.3 | 51.7 | 51.4 | 50.4 | 50.0 |
| FreqNet | **88.9** | 65.6 | 71.9 | 59.0 | 59.0 | 71.8 | 97.4 | **97.2** | 97.8 | 84.4 | 86.6 | **91.9** | **97.2** |
| UFD | 66.4 | 64.5 | 58.0 | 55.6 | 68.3 | 70.3 | 94.8 | 74.8 | 95.6 | 78.5 | 79.2 | 71.5 | 88.3 |
| RINE | 75.9 | 79.7 | 64.2 | 84.1 | 91.6 | **81.5** | 96.9 | 82.5 | 96.7 | 77.5 | 79.2 | 85.6 | 93.7 |
| Ours (SVM$_9$) | 56.8 | **89.7** | 52.7 | 93.1 | 94.7 | 73.1 | **99.1** | 86.4 | **99.0** | 88.1 | 90.2 | 90.9 | 96.1 |
| Ours (MLP$_9$) | 61 | 84.1 | 52.7 | 86.6 | 86.8 | 72 | 97.9 | 84 | 97.9 | 86.6 | 90.6 | 90.2 | 96.3 |
| Ours (SVM$_0$) | 65.5 | 71 | 51.1 | 50.5 | 51 | 81.5 | 97 | 95 | 99 | 81.5 | 87 | 82.5 | 92 |
| Ours (MLP$_0$) | 84.8 | 94.5 | 61.5 | 50.7 | 50.7 | 81.0 | 95.4 | 95.3 | 95.2 | 76.3 | 83.2 | 78.0 | 94.9 |
| Ours (SVM$_{max}$) | 58.8 | 79.5 | 50.4 | 69 | 88.5 | 64 | 93.5 | 76.5 | 96.5 | 76.5 | 74.5 | 77.5 | 85 |
| Ours (MLP$_{max}$) | 61.5 | 87 | 62 | 87 | 97 | 71 | 98.5 | 83 | 99 | 87 | 87 | 86.5 | 91 |
| CNNDet1 | 89.0 | 73.7 | 59.5 | 98.2 | 98.4 | 73.7 | 70.6 | 71.0 | 70.5 | 80.6 | 84.9 | 82.1 | 70.6 |
| CNNDet2 | 66.3 | 86.0 | 61.2 | **98.9** | 99.5 | 68.6 | 66.0 | 66.7 | 65.4 | 73.3 | 78.0 | 76.2 | 65.9 |
| PatchDet1 | 60.2 | 65.8 | 52.9 | 68.7 | 67.6 | 70.0 | 87.8 | 84.9 | 88.1 | 74.5 | 76.3 | 75.8 | 77.1 |
| PatchDet2 | 76.5 | 76.2 | 76.3 | 74.5 | 68.5 | 75.0 | 87.1 | 86.7 | 86.4 | 85.4 | 83.7 | 78.4 | 75.7 |
| CO | 59.1 | 69.0 | 60.4 | 73.1 | 87.2 | 70.2 | 91.2 | 89.0 | 92.4 | 89.3 | 88.3 | 82.8 | 81.0 |
| FreqSpec | 45.2 | 47.5 | 57.1 | 53.6 | 51.0 | 57.7 | 77.7 | 77.3 | 76.5 | 68.6 | 64.6 | 61.9 | 67.8 |
| FreqNet | 94.4 | 62.3 | 80.1 | 74.6 | 77.8 | 90.7 | 99.9 | **99.9** | 99.9 | 84.4 | 86.6 | 94.7 | 99.7 |
| UFD | 81.8 | 66.7 | 79.3 | 96.7 | 98.4 | 87.7 | 99.4 | 93.5 | 99.2 | 95.6 | 95.8 | 92.6 | 97.9 |
| RINE | **96.6** | 91.6 | **85** | 92.8 | 99.6 | **97.1** | 99.6 | 96.4 | 99.6 | 95.5 | 96.5 | 92.3 | 99.4 |
| Ours (SVM$_9$) | 94 | **96.8** | 66.8 | 98.4 | **99.8** | 95.1 | **100** | 99.7 | **100** | 99.6 | **99.7** | 99 | **99.9** |
| Ours (MLP$_9$) | 95.7 | 94.9 | 67 | 98.7 | 99.8 | 93.3 | 99.9 | 99.5 | 99.9 | 99.4 | 99.7 | 99.4 | 99.9 |
| Ours (SVM$_0$) | 96.5 | 97.1 | 53 | 64.7 | 99.9 | 89.5 | 99.9 | 99.8 | 100 | 98 | 99.6 | 98.7 | 99.5 |
| Ours (MLP$_0$) | 97.7 | 99.3 | 70.9 | 95.1 | 100 | 91.5 | 99.8 | 99.8 | 99.8 | 87.9 | 93.0 | 88.8 | 99.6 |
| Ours (SVM$_{max}$) | 95.2 | 97.6 | 64.8 | 99 | 99.8 | 96.6 | 99.9 | 99.8 | 100 | 99.7 | 98.7 | 98.6 | 99.2 |
| Ours (MLP$_{max}$) | 94.8 | 96.9 | 66.9 | 99.1 | 99.7 | 96.5 | 99.9 | 99.8 | 100 | 99.7 | 99.4 | 99 | 99.3 |

## 6.1 Standard visual deepfake detection benchmark

The standard visual deepfake detection benchmark results are in Tabs. 1 and 2. In addition to the considered baselines, we experiment with several variants of our approach, denoted by Ours (CLS$_k$) where CLS is the classifier (MLP or SVM) and $k$ is the symmetric range parameter (see Sec. 5). For instance, Ours (SVM$_{max}$) represents our approach, trained using an SVM classifier with the maximum $k = l/2$. Tab. 1 details the accuracy and mean precision measures in the format ACC / mAP for GAN-based models. Diffusion and autoregressive methods are considered in Tab. 2, where the top part details ACC, and the bottom part shows the mAP. On GAN data, we find FreqNet to produce strong results, except on WFIR. UFD attains good scores on most GAN approaches, however, its effectiveness reduces on StyleGAN2 and WFIR. Surprisingly, CNNDet1 is somewhat robust across all datasets, however, it is also sensitive to the choice of compression parameter, cf. CNNDet1 and CNNDet2. We observe that most deepfake detectors struggle to generalize to

diffusion-based and autoregressive generators, as shown in Tab. 2. In contrast, our approach demonstrates strong and robust ACC and mAP scores across all generators, rivaling the latest state-of-the-art method, RINE, while utilizing fewer parameters and a linear classifier. We conclude this experiment by evaluating the overall ACC and mAP averages of our best-performing model, Ours ($SVM_9$), across all datasets. These results, summarized in Tab. 3, demonstrate that our approach outperforms RINE in terms of accuracy (ACC) while achieving comparable performance in mean average precision (mAP).

Table 3: Averages of ACC and mAP for all considered detection methods in the visual benchmark.

| Metric | CNNDet1 | CNNDet2 | PatchDet1 | PatchDet2 | CO | FreqSpec | FreqNet | UFD | RINE | Ours |
|--------|---------|---------|-----------|-----------|-------|----------|---------|-------|-------|----------------|
| ACC | 71.02 | 64.93 | 69.46 | 72.07 | 66.88 | 55.5 | 83.94 | 80.98 | 88.7 | $\mathbf{89 \pm 0.051}$ |
| mAP | 84.78 | 82.65 | 74.41 | 72.95 | 78.11 | 66.22 | 89.87 | 94.05 | **97.7** | $97.6 \pm 0.004$ |

## 6.2 Audio deepfake detection benchmark

The audio detection setting is similar to the image-based approach. Following Müller et al. (2022), we train our classifier exclusively on the ASVSpoof2019 dataset, while testing is conducted on both the ASVSpoof2019 test set and the In-The-Wild dataset. Our classifier is built using the ImageBind backbone (Girdhar et al., 2023), leveraging publicly available weights from HuggingFace. The results, presented in Tab. 4 (left), demonstrate strong performance on ASVSpoof2019 and state-of-the-art results on In-The-Wild. Our findings suggest that the superior performance on In-The-Wild is due to the reduced overfitting compared to competing methods, which tend to overfit the ASVSpoof2019 training set. Furthermore, our approach requires significantly fewer computational resources, including reduced training time, compared to other methods. Finally, we evaluate our method in a few-shot setting, utilizing only 200 samples (approximately 1% of ASVSpoof2019) while maintaining the original dataset's label ratio. This scenario better reflects real-world conditions. The results, shown in Tab. 4 (right), highlight the effectiveness of our approach in the few-shot context, achieving notable improvements over RawGAT-ST (Tak et al., 2021b).

Table 4: Full-shot (left) and few-shot (right) EER values for In-the-Wild and ASVSpoof2019 datasets.

| Model | Full-shot | | | | | Few-shot | |
|-------|-----------|--------|---------|-----------|-----------------|-----------|-----------|
| | LCNN | Rawpc | RawNet2 | RawGAT-ST | ImageBind (k=4) | RawGAT-ST | ImageBind |
| ASVSpoof2019 (test) | 6.354 | 3.15 | 3.1 | **1.23** | 3.4 | 15.44 | **6.01** |
| In-The-Wild | 65.56 | 45.71 | 37.82 | 37.15 | **34.35** | 46 | **30.04** |

## 6.3 Clustering-based image deepfake detection

While training on real and fake information from a single source, and testing on multiple generative sources as detailed in Sec. 6.1 is considered the standard benchmark in the community, we propose below a new benchmark for detectors that are based on latent features. The key idea behind the proposed benchmark is to exploit the clustering properties of latent representations. Intuitively, if data representations of the same class, e.g., fake images, concentrate in a specific sub-domain of the entire space, then, the whole cluster can be classified together in a straightforward fashion. For this benchmark, we compare UFD (Ojha et al., 2023) that uses only the last layer to our method with only the middle layer, i.e. Ours ($SVM_0$). However, any other detection technique can be also considered in the context of this benchmark, if it produces representations from which (quasi-)linear classification is performed.

The experimental setup of our benchmark is constructed as follows. We sub-sample 200 examples from ProGAN, and compute their latent representations with UFD and our method. Given these features, we train a single SVM classifier (Cortes & Vapnik, 1995) per method to convergence, and we use it later for identification. Then, for every generative model and its associated real and fake datasets, we compute their features. Importantly, we pre-process these features by applying dimensionality reduction. This is done to address some of the challenges in estimating Euclidean distances in high-dimensional spaces, which can distort true data relationships (Aggarwal et al., 2001). Using the lower-dimensional features, we cluster that

information with k-means where $k = 2$ (Lloyd, 1982). Finally, we classify the features with the pre-trained SVM *per cluster*, and we employ a majority vote to determine the clusters' labels. For instance, if one cluster is 60% real, and the other cluster is 51% fake, then we determine clustering based on the first cluster, setting the labels of all images in the first cluster to be 'real', and in the second cluster to be 'fake'. Notice that this choice of labeling is robust to pathological cases where both clusters attain the same label from SVM.

Our clustering benchmark is different from the standard benchmark in Sec. 6.1 in a few aspects worth mentioning. First, in the standard setting, we do not assume that we have access to the entire test set, and here, we require the data to compute clustering. While our additional constraint here is somewhat limiting, we believe that it is also practical in many scenarios where a collection of media needs to be classified. Second, the standard benchmark uses the raw CLIP-ViT features, and in contrast, we reduce the dimensionality of the features to facilitate clustering. Finally, labeling in the standard case is done on a per-sample basis, whereas here, the entire cluster is labeled together. This means that the clustering benchmark depends on the implicit assumption that same-class images are clustered together and are separable from other classes of images. Overall, we believe that our new benchmark sheds a new perspective on the features of pre-trained multi-modal models.

We detail the results of UFD and our approach for the above experiment in Tabs. 5 and 6 for GAN-based tools and diffusion models, respectively. The reported results represent the accuracy measures. The mAP can not be measured fairly in this case, as there is no clear threshold. While studying the results, we observe that UFD often yields $\approx 50\%$, which is the random baseline, implying that it erroneously labels both clusters to be 'real' (or 'fake'). This observation may be justified by re-considering the 2D t-SNE point clouds plotted in Figs. 1 and 6, showing entangled features at the last layer. In comparison, our approach attains almost perfect results in many of the datasets, yielding weak classifications only for Deep fakes and StyleGAN. Nevertheless, our approach is better than UFD in all the considered generative models. In particular, UFD obtains mean accuracy of 59.12% and 74.62% for GAN and diffusion models, respectively. In comparison, our technique achieves 93% and 91.15%. In total, UFD has a mean accuracy of 68.72%, whereas we get 91.85%.

Table 5: Accuracy results on GAN-based generative models in the clustering benchmark.

| Method | ProGAN | CycleGAN | BigGAN | GauGAN | StarGAN | StyleGAN | StyleGAN2 | WFIR |
|---|---|---|---|---|---|---|---|---|
| UFD | 76% | 50% | 50% | 51% | 99% | 46% | 49% | 52% |
| Ours | **99%** | **99%** | **99%** | **99%** | **100%** | **64%** | **87%** | **97%** |

Table 6: Accuracy results on diffusion and non-GAN models in the clustering benchmark.

| Method | Deep fakes | Low level vision | | Perceptual loss | | Guided | LDM | | | Glide | | | DALL-E |
|---|---|---|---|---|---|---|---|---|---|---|---|---|---|
| | | SITD | SAN | CRN | IMLE | | 200 steps | 200 w/ CFG | 100 steps | 100 27 | 50 27 | 100 10 | |
| UFD | 47% | 48% | 50% | 99% | 99% | 49% | 91% | 49% | 90% | 91% | 92% | 92% | 73% |
| Ours | **50%** | **68%** | **73%** | **99%** | **99%** | **96%** | **100%** | **100%** | **100%** | **100%** | **100%** | **100%** | **100%** |

## 6.4 Detecting the source of the generative model

In the pioneering works Zhang et al. (2019); Frank et al. (2020), the authors identify spectral fingerprints shared across several GAN-based models. However, subsequent works observed that synthetic media generated with diffusion models demonstrate significantly weaker spectral traces in comparison to GANs (Ricker et al., 2022; Ojha et al., 2023). In this context, there is an overarching question underlying our research: Do generated images contain latent features different from natural images? If so, are these latent features different for different generative models? While our study above mostly focused on the first question, the following experiment aims to address the latter question. Towards that end, we utilize the latent representations of CLIP-ViT to identify the *source* of the generative model.

While it is crucial to distinguish between real and fake content, we believe that identifying the particular generative model that generated the data can be instrumental in several scenarios. For instance, tools for source identification can be employed in copyright disputes. Another example is related to improving the accuracy of a specific deepfake detection approach, e.g., DIRE-D excels in detecting diffusion generated

media (Wang et al., 2023). To adapt our approach to allow for source identification, we perform the following. First, we extract the latent features of a certain layer of CLIP-ViT for the datasets: BigGAN, GauGAN, DALL-E, Guided, StyleGAN, CycleGAN, StyleGAN2, StarGAN, Glide, Stable Diffusion (Rombach et al., 2022), Midjourney, and MidjourneyV5 (mid; Wu et al., 2023). Each dataset contains 500 images. These datasets encompass a combination of state-of-the-art GAN and diffusion models, providing a comprehensive evaluation framework. Second, we randomly sample a minimal set of 10 images from each dataset. Then, we use the sampled extracted features (120 in total) to train a single SVM. Finally, we test the remaining 490 images per method on the trained SVM. We repeat this experiment for UFD and for our approach with k=0.

Tab. 7 shows the results for the source identification experiment in a confusion matrix. The generative models listed on the left are the true sources, whereas the top models represent the predicted classes. The values are in percentages, given in the format $a$ / $b$, where $a$ is the accuracy of UFD and $b$ is our accuracy measure. Ideally, if the main diagonal of the confusion matrix shows 100 in all its cells, then the classification is perfect. We denote in bold the values along the main diagonal that are best per generative model. These results indicate that UFD performs relatively well on GAN approaches, but it struggles with most diffusion-based media. Remarkably, our approach yields high scores in most cases, except for some confusion between Midjourney and MidjourneyV5. Further, our classification results are better than UFD in all cases, often by large margins, StyleGAN excluded. We conclude from this experiment that CLIP-ViT has the ability to embed different generative models separately in its latent representations, allowing to robustly identify the source of the generated content.

Table 7: We show the confusion matrix with the results for the source identification experiment comparing UFD (left) and ours (right). Values represent percentages.

| | BigGAN | GauGAN | StyleGAN | CycleGAN | StyleGAN2 | StarGAN | DALL-E | Guided | MidjournyV5 | Midjourny | Glide | Stable Diffusion |
|---|---|---|---|---|---|---|---|---|---|---|---|---|
| BigGAN | 53 / **95** | 10 / 1 | 7 / 1 | 3 / 0 | 3 / 0 | 3 / 1 | 8 / 2 | 7 / 0 | 0 / 0 | 1 / 0 | 4 / 0 | 1 / 0 |
| GauGAN | 8 / 0 | 84 / **99** | 1 / 0 | 4 / 0 | 0 / 0 | 0 / 0 | 1 / 1 | 2 / 0 | 0 / 0 | 0 / 0 | 0 / 0 | 0 / 0 |
| StyleGAN | 0 / 1 | 0 / 0 | **90** / 86 | 0 / 1 | 10 / 11 | 0 / 0 | 0 / 0 | 0 / 1 | 0 / 0 | 0 / 0 | 0 / 0 | 0 / 0 |
| CycleGAN | 5 / 0 | 2 / 0 | 0 / 0 | 91 / **99** | 0 / 0 | 0 / 1 | 0 / 0 | 1 / 0 | 0 / 0 | 0 / 0 | 0 / 0 | 0 / 0 |
| StyleGAN2 | 0 / 0 | 0 / 0 | 18 / 4 | 0 / 0 | 82 / **95** | 0 / 0 | 0 / 0 | 0 / 1 | 0 / 0 | 0 / 0 | 0 / 0 | 0 / 0 |
| StarGAN | 0 / 0 | 0 / 0 | 0 / 0 | 0 / 0 | 0 / 0 | **100 / 100** | 0 / 0 | 0 / 0 | 0 / 0 | 0 / 0 | 0 / 0 | 0 / 0 |
| DALL-E | 9 / 0 | 10 / 0 | 0 / 0 | 1 / 0 | 0 / 0 | 1 / 0 | 66 / **100** | 6 / 0 | 0 / 0 | 1 / 0 | 3 / 0 | 4 / 0 |
| Guided | 2 / 0 | 2 / 0 | 0 / 3 | 1 / 0 | 7 / 0 | 0 / 0 | 3 / 2 | 73 / **89** | 1 / 0 | 0 / 0 | 10 / 6 | 1 / 0 |
| MidjournyV5 | 0 / 0 | 0 / 0 | 0 / 0 | 0 / 0 | 0 / 0 | 0 / 0 | 0 / 0 | 1 / 0 | 63 / **75** | 22 / 5 | 0 / 0 | 12 / 20 |
| Midjourny | 0 / 0 | 1 / 0 | 0 / 2 | 0 / 0 | 0 / 0 | 0 / 0 | 1 / 0 | 0 / 0 | 35 / 35 | 48 / **49** | 0 / 0 | 15 / 14 |
| Glide | 2 / 0 | 0 / 0 | 0 / 0 | 1 / 0 | 1 / 0 | 0 / 0 | 1 / 0 | 8 / 4 | 0 / 0 | 0 / 0 | 87 / **96** | 0 / 0 |
| Stable Diffusion | 0 / 0 | 0 / 0 | 0 / 0 | 0 / 0 | 0 / 0 | 0 / 0 | 1 / 3 | 0 / 4 | 17 / 23 | 20 / 7 | 1 / 0 | 59 / **63** |

# 7 Discussion

The recent rise of high-quality generative models for visual content raises the concern of malicious users using such data for spreading misinformation and disseminating deepfakes, motivating the development of automatic detection tools for synthetic media. One of the main challenges is that existing generative models vary in their underlying technologies and fundamental theories, and thus, there is a need for developing universal detectors, effective across multiple generative tools, families, and modalities. In this work, we analyze the inner representations of large vision-language and other large-multi-modal models. Our analysis shows that intermediate layers provide robust features for detection, allowing to classify real and fake images using a simple linear classifier. We extensively evaluate our approach on standard benchmarks and in comparison to recent state-of-the-art works. In addition, we also show that our approach allows detection via clustering.

One limitation of our method, shared with many other deepfake detectors, is its robustness to noise. Presumably, modifying the original synthetic images results in losing the latent fingerprints enabling their detection. We would like to further investigate this aspect in future work. More generally, our method raises several intriguing questions: What underlying patterns do intermediate representations capture that facilitate accurate source identification? Do the middle layers of contrastive multi-modal models retain critical generative characteristics better than the final layer, which may focus more on high-level semantics? Can we

distinguish between diffusion models and GAN? We would like to focus in the future on addressing these questions, towards the development of better deepfake automatic techniques.

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

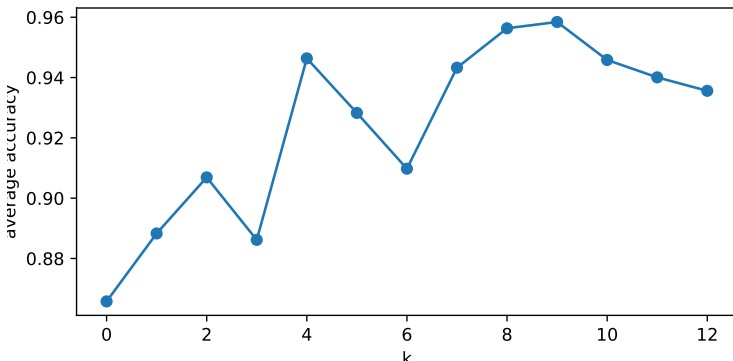

Figure 5: Average accuracy on the ProGAN validation set for different k values with CLIP-ViT as backbone. The validation set is augmented with jpeg-compression, and the model is trained on 100 randomly sampled instances from the training set. The reported scores represent the mean accuracy over 10 runs

## A    Choosing $k$

The selection of $k = 9$ is based on prior findings by Gandelsman et al. (2023), which demonstrate that the last four layers of CLIP primarily capture high-level semantic information. Since such information is less beneficial for our task, we discard these layers. To maintain consistency and simplicity in our approach, we frame our selection symmetrically around the middle layer, leading to the $k = 9$ setting. However, to enhance robustness, we propose a data-driven approach for selecting $k$.

A standard approach for selecting discrete hyperparameters involves performing a grid search over the hyperparameter space and choosing the setting that performs best on validation data. However, in our case, models typically perform well on their training distribution but generalize poorly to other generative models. Selecting $k$ based solely on validation performance within the training set would therefore be misleading. For instance, as shown in Table 1, all models achieve strong performance on the ProGAN test split but perform significantly worse on other generative models.

To address this issue, we propose training on only a small subset of the available training data while validating on a larger, compressed-augmented dataset that better approximates unseen generative distributions. This approach helps reduce the risk of overfitting to the given training distribution. We report the results for different $k$ values for CLIP-ViT on the ProGAN augment-validation set in Fig 5. Our findings indicate that the best performance for CLIP-ViT is achieved for $7 \leq k \leq 10$.

## B    Multi modal models

**CLIP-ViT.**    Instead of training biased classifiers, recent work suggested to utilize the feature space learned by CLIP-ViT models (Dosovitskiy et al., 2021; Radford et al., 2021), trained on internet-scale image-text pairs (Ojha et al., 2023; Koutlis & Papadopoulos, 2024). We briefly recall the main components of CLIP-ViT, essential for describing our approach, following the notation in Gandelsman et al. (2023). CLIP embeds images $I$ and texts $t$ within the same latent space by optimizing the cosine similarity between $M_{\text{img}}(I)$ and $M_{\text{txt}}(t)$, where $M_{\text{img}}(\cdot)$ and $M_{\text{txt}}(\cdot)$ are image and text encoders, respectively. CLIP represents an image $I \in \mathbb{R}^{H \times W \times 3}$ using an encoder $M_{\text{img}}$ that projects the $d$ features learned from the image $I$ by a vision transformer (ViT) (Dosovitskiy et al., 2021), namely,

$$M_{\text{img}}(I) := P \, \text{ViT}(I) \,, \quad P \in \mathbb{R}^{d' \times d} \,. \tag{3}$$

ViT is a residual network of $L$ layers, each includes a multi-head self-attention (MSA) followed by a multi-layer perceptron (MLP) block. In practice, the image $I$ is decomposed into patches, projected to tokens $z_1^0, \ldots z_N^0$ that form the columns of $Z^0 \in \mathbb{R}^{d \times (N+1)}$, where the first column is the class token $z_0^0$. Given $Z^0$, ViT

performs the following residual stream

$$\hat{Z}^l = \mathrm{MSA}^l(Z^{l-1}) + Z^{l-1} \ , \quad Z^l = \mathrm{MLP}^l(\hat{Z}^l) + \hat{Z}^l \ . \tag{4}$$

Finally, the output of ViT is the first column of $Z^L$, corresponding to the class token, $[Z^L]_{\mathrm{cls}}$.

**ImageBind.** ImageBind learns a joint embedding space across six modalities: images, text, audio, depth, thermal, and IMU data (Girdhar et al., 2023). Unlike models requiring all combinations of paired data, ImageBind aligns each modality to images, leveraging naturally occurring image-paired data. This approach enables zero-shot capabilities across modalities without explicit pairing between all modalities. For instance, by aligning text and audio embeddings to image embeddings, ImageBind facilitates cross-modal retrieval tasks, such as retrieving an audio clip from text. ImageBind encodes each modality separately with a transformer and employs contrastive learning to align modality-specific embeddings with image embeddings, resulting in a unified representation space that supports various emergent applications across different modalities.

## C  Noise robustness

Noise robustness is a critical property for ensuring model performance and reliability in real-world applications where data is often imperfect or corrupted. Specifically, we modify the datasets considered in Sec. 6.1 with an additive Gaussian noise with zero mean and variances $\sigma = 1, 2$. A similar test is also performed in previous work Ojha et al. (2023). We utilize the same experimental setup and evaluation as discussed in Sec. 6.1, with k=9. The results are presented in Tabs. 8 and 9 for GAN- and diffusion-based methods. The tables are organized in a similar format to Tabs. 1, 2. To ease readability, we place competing approaches in subsequent rows, e.g., UFD, $\sigma = 1$ and Ours, $\sigma = 1$ detail the results for the Gaussian noise experiment for UFD and our method, respectively. We also list the overall ACC and mAP averages in Tab. 10. While both approaches attain lower measures for noisy images, we find our approach to be more resilient in comparison to UFD.

Table 8: Ablation results for noisy images on GAN content.

| Method | ProGAN | CycleGAN | BigGAN | GauGAN | StarGAN | StyleGAN | StyleGAN2 | WFIR |
|---|---|---|---|---|---|---|---|---|
| UFD, $\sigma = 1$ | 98.8 / 99.9 | 95.1 / 98.9 | 76.8 / 91.6 | 97.7 / 99.7 | 87.5 / 93.2 | 74.5 / 94.0 | 62.8 / 90.3 | 54.8 / 63.2 |
| Ours, $\sigma = 1$ | 99.2 / 100 | 98.6 / 100 | 91.8 / 98.8 | 99.2 / 100 | 98.6 / 100 | 74.2 / 99.3 | 61.3 / 98.0 | 97 / 99.9 |
| UFD, $\sigma = 2$ | 94.3 / 98.9 | 85.9 / 94.1 | 68.3 / 80.6 | 91.1 / 97.4 | 73.0 / 85.3 | 63.8 / 82.1 | 60.1 / 77.0 | 50.3 / 51.7 |
| Ours, $\sigma = 2$ | 96.8 / 99.8 | 95.3 / 99.1 | 75.5 / 92.6 | 97.8 / 99.8 | 82.2 / 99.7 | 62.1 / 93.6 | 54.4 / 90.4 | 94.7 / 99.8 |

Table 9: Ablation results for noisy images on diffusion content.

| Method | Deep fakes | Low level vision | | Perceptual loss | | Guided | LDM | | | Glide | | | DALL-E |
|---|---|---|---|---|---|---|---|---|---|---|---|---|---|
| | | SITD | SAN | CRN | IMLE | | 200 steps | 200 w/ CFG | 100 steps | 100 27 | 50 27 | 100 10 | |
| UFD, $\sigma = 1$ | 54.7 | 64.0 | 54.0 | 54.0 | 59.8 | 67.8 | 86.6 | 61.9 | 86.3 | 77.3 | 76.8 | 68.0 | 72.5 |
| Ours, $\sigma = 1$ | 51.3 | 90.0 | 50.0 | 89.2 | 91.2 | 71.6 | 94.1 | 69.3 | 94.3 | 81.05 | 83.9 | 79.4 | 90.4 |
| UFD, $\sigma = 2$ | 53.3 | 61.5 | 51.5 | 54.1 | 61.5 | 66.5 | 76.3 | 56.1 | 77.2 | 70.7 | 72.9 | 57.5 | 62.7 |
| Ours, $\sigma = 2$ | 51.3 | 89.0 | 49.2 | 77.8 | 88.6 | 61 | 77.1 | 57 | 79.6 | 68.2 | 69.5 | 68.9 | 71.2 |
| UFD, $\sigma = 1$ | 71.9 | 69.5 | 64.9 | 85.4 | 91.5 | 83.1 | 96.9 | 81.9 | 96.8 | 93.5 | 93.5 | 86.4 | 90.1 |
| Ours, $\sigma = 1$ | 81.0 | 96.4 | 53.4 | 96.36 | 98.1 | 90.02 | 99.7 | 97.1 | 99.7 | 98.5 | 99.2 | 98. | 99.4 |
| UFD, $\sigma = 2$ | 69.3 | 68.9 | 53.5 | 72.8 | 85.1 | 74.9 | 89.4 | 66.6 | 89.5 | 85.6 | 86.2 | 72.4 | 76.8 |
| Ours, $\sigma = 2$ | 74.5 | 96.2 | 48 | 88.2 | 95.5 | 75.5 | 95.9 | 85.6 | 97 | 90. | 92.2 | 90.1 | 93.2 |

Table 10: Mean ACC and mAP results for all datasets considered in our noise study.

| Metric | UFD, $\sigma = 1$ | Ours, $\sigma = 1$ | UFD, $\sigma = 2$ | Ours, $\sigma = 2$ |
|---|---|---|---|---|
| ACC | 72.94 | **83.68** | 67.08 | **74.7** |
| mAP | 87.44 | **95.41** | 78.96 | **90.03** |

## D    Additional Results

In Fig. 6, we present an extended version of Fig. 1, where we show t-SNE embeddings of layers of CLIP-ViT across several generative techniques. Overall, there is a clear trend showing entangled clusters in the first and last layers. In contrast, the intermediate layers can be clustered linearly.

## E    Training details

All of our models were trained for one epoch on an NVIDIA RTX6000 GPU. We used publicly available implementation of CLIP [1], ImageBind [2] and scikit-learn  (Pedregosa et al., 2011) implementation for the SVM and MLP results. We used the same data augmentation of blurring, JPEG compression and crop-resize following Wang et al. (2020a), Ojha et al. (2023). The threshold for the classification was set to 0.5. In our experiment at Sec. 6.3, we used scikit-learn implementation of k-means with $k = 2$ and default settings. In addition, we employed the implementation of scikit-learn for t-SNE (Van der Maaten & Hinton, 2008) with default parameters, reducing the dimensionality to 100.

---

[1] https://github.com/openai/CLIP
[2] https://github.com/facebookresearch/ImageBind

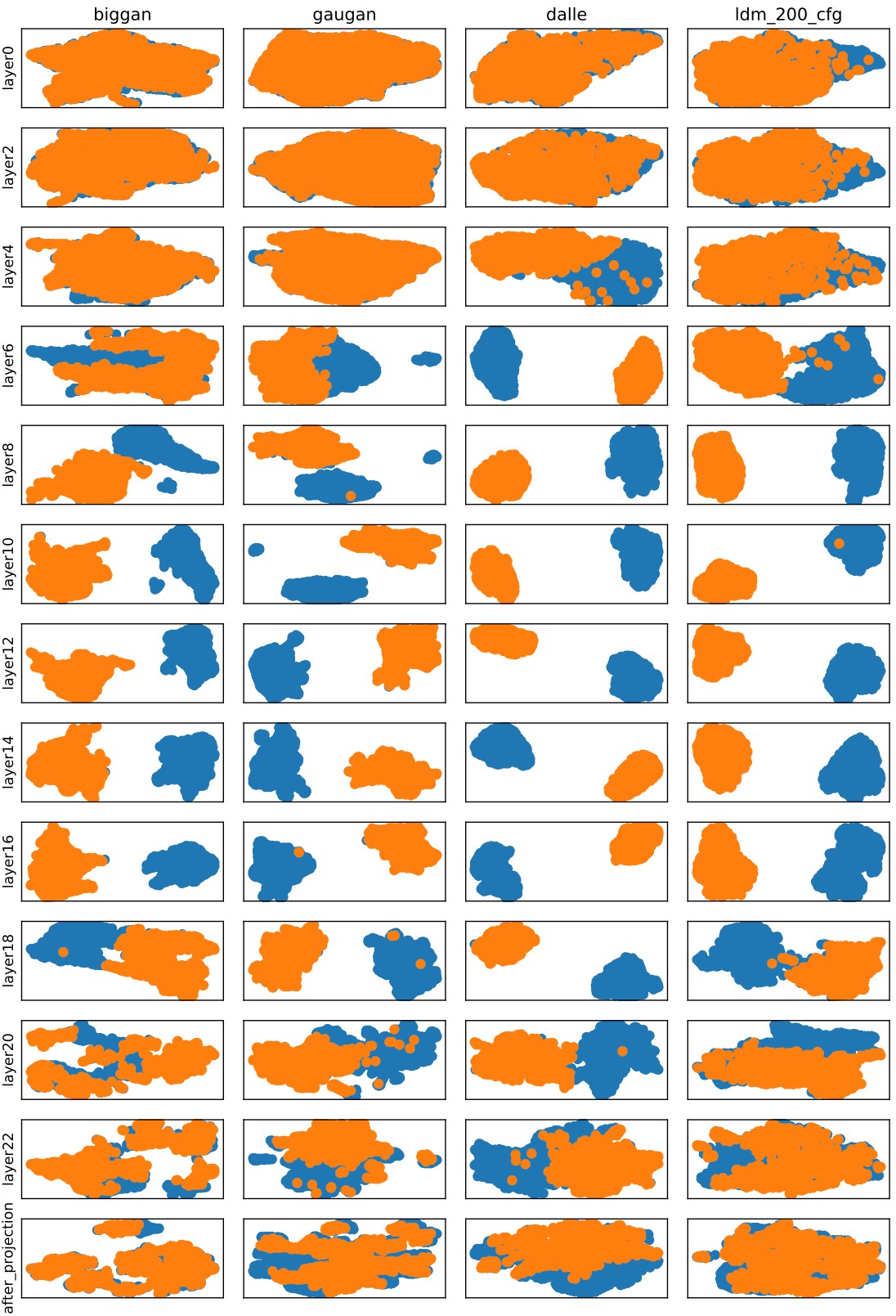

Figure 6: We present t-SNE embeddings of select layers of CLIP-ViT and their features on several GAN- and diffusion-based tools. 'after_projection' is the final layer used by UFD Ojha et al. (2023).

