# OpenReview forum: "Unraveling Hidden Representations: A Multi-Modal Layer Analysis for Better Synthetic Content Forensics"
_TMLR — Rejected by TMLR_

### Review · Reviewer_a9mT · 2025-02-07

**Summary Of Contributions:**

The paper uses multi-modal foundation models for distinguishing between the fake and real inputs in different modalities (e.g., images or audio).

**Audience:**

No

**Broader Impact Concerns:**

No broader impact concerns

**Claims And Evidence:**

No

**Requested Changes:**

Lack of actionable items:  address the questions on Interest to the audience related to the actionable items.

Claims and evidence: address all the problems as described in the claims and evidence section of the review above.

**Strengths And Weaknesses:**

Interest to the audience:

Pros:
In general, the topic has an audience as better fake detectors are of high importance and I understand that new insights into their architecture would have audience.

Cons:
The main issue on this front is the lack of actionable items: what did the authors find that helps towards building better (w.r.t. some criterion, e.g., accuracy or precision) deepfake classifiers? It would be improved if the authors answered the following questions: is there any rigorous method for choosing the layers to detect deepfakes with? What architectural decisions improve the detection (e.g., see Question 1, which asks if the choice of multimodal architecture contributes towards better detection as opposed to its unimodal ablation)?

Claims and evidence:

Pros:
The authors claim that  they 'show that the latent code of these models naturally captures information discriminating real from fake’ which appears to be covered in the paper.

Cons:

1. “such tools typically generalize only within the same family of generators and data modalities, yielding poor results on other generative classes and data domains.” and “Towards a “universal” classifier, we propose the use of large pre-trained multi-modal models for the detection of generative content. “ It is not clear how do the authors confirm that the proposed solution brings us towards the universal classifer more than a collection of well-trained single-mode classifiers? The universality could be considered in two dimensions: (1) generalisation to the methods (and the proposed method, as shown in the experimental section, Table 1-6, does not show such advantages) (2) generalisation to the domains (audio/image): it is not clear that the unimodal classifiers with the same architecture would not be competitive on these tasks

2. Furthermore, given that it’s only a small difference between the values, the authors need to present confidence intervals.

3.  The authors provide the experimental results only for the visual and audio modalities, which does not show the advantages of video and text modalities. Furthermore, in the architecture in Figure 4, it is unclear how exactly the authors would implement embedding videos and images into the same feature space given that the fundamental difference between them is that videos have time dimension and the images do not. For example, if the authors distill the video encoder into the image encoder, wouldn’t it erase the time information from the video? In this case, the procedure for building the joint feature space would be critical to the efficiency of the method, so the authors need to justify the design of contrastive loss. If the authors do not actually tackle the video processing, maybe it would be better to leave only the modalities the authors consider, i.e. audio and images?

4. Finally, I cannot see the training details. The authors say that the method is illustrated in Figure 5 (does it exist)? I guess the text refers in fact to Figure 4 but it does not contain any references to the details of training.


Questions:

(1)  The authors claim that 'Effectively, we show that the latent code of these models naturally captures information discriminating real from fake’ It is not clear though whether this is because of the multimodality the authors use in the proposed method, or is it because of the better performance of single modalities (e.g., better performance of audio encoder, image encoder etc)?

(2) The multi-modal detectors can be used in two different ways: (1) detection of fake inputs within a multimodal signal (e.g. video + sound) taking advantage of joint representation (2) one-model-fits-all approach: the same model allows to work in different modalities, i.e. it can work with audio or with text or with video. It seems like the authors address the second scenario, which makes the reviewer wonder if there is any evidence of benefit for the first scenario.

---

> ### Author Response · Authors · 2025-02-25
>
> We sincerely thank Reviewer a9mT  for their comprehensive and constructive review. We have addressed each point in detail below:
>
> >  It is not clear how do the authors confirm ... universal classifer more than a collection ... single-mode classifiers? ... with the same architecture would not be competitive on these tasks
>
> We believe there might be a misunderstanding of the particular challenge we consider in our work and we would like to elaborate on it. The standard setting is as follows: given a dataset generated by model Y , we train a classifier and evaluate its performance on datasets generated by models different from  Y . This setup reflects real-world challenges, as new generative models emerge frequently, necessitating the ability to generalize to unseen methods.
> Generally, classifiers that were trained on Y, usually perform well on unseen data from Y and worse on unseen data different from Y (CNNDet results on ProGAN test on Tab.1  & diffusion datasets on Tab.2.). The universality of our approach is validated by our main results in Tab. 1 & Tab. 2, where although our method trained only on of ProGAN, it generalizes to unseen images obtained by other techniques, while using the same classifier. Furthermore, our approach is also universal to the type of modality, since we show that the same network produces robust features for images & audio.
> Our discussion above highlights that ``well-trained single-mode classifiers'' are irrelevant to the setup we consider, since we do not have access to a train set per method (assuming that the reviewer meant to train a separate classifier per generation method). If the reviewer was referring to uni-modal networks, we did not explore them, some may exhibit properties similar to universal multi-modal models.
>
> > what did the authors find that helps ...  is there any rigorous method for choosing the layers ...?
>
> We agree that a more structured approach would strengthen our claims. We propose the following methodology: We use an augmented-jpeg-compress variation of the validation dataset, while training on a small sample size, choosing best performing k. The augmentation is for better approximate different distribution from the train set. Results show that 7<=k<=10 is optimal. Further details are on appendix A.
>
> >  ... the authors need to present confidence intervals.
>
> We report confidence intervals for our main results (SVM9). Specifically, we report results along with confidence intervals computed over five different runs. We include this result in the revised version.
>
> > The authors provide the experimental results only for the visual & audio modalities, which does not show the advantages of video and text modalities. Furthermore, ... , it is unclear how exactly the authors would implement embedding videos and images into the same feature ... leave only the modalities the authors consider...?
>
> While our experiments focus on the visual and audio modalities, our approach can be extended to video and text, as demonstrated by prior works such as ImageBind & LanguageBind. These methods have successfully addressed the challenge of embedding videos and images into a shared feature space.
> LanguageBind, for example, utilizes temporal attention mechanisms [1,2] to encode temporal information effectively. These strategies suggest that it is feasible to apply our method to video data without fundamental changes. [1] LanguageBind: Extending Video-Language Pretraining to N-modality by Language-based Semantic Alignment [2] VidTr: Video Transformer Without Convolutions.
>
> > I cannot see the training details. ...
>
> We have added training details at Section 5 in the appendix of the revised version.
>
> > The authors claim that 'Effectively, we show that the latent code ....  of the multimodality the authors use in the proposed method, or is it because of the better performance of single modalities ...?
>
> We have not conducted a direct "apples-to-apples" comparison isolating multimodality. However, results in Tab.1 and Tab.2 suggest that multimodal training improves encoder effectiveness for deepfake detection. The top-performing methods—our approach and RINE—use CLIP ViT features, while others rely on single-modality representations (e.g., CNNDet, FreqNet). This indicates that multimodal encoders enhance generalization, though further experiments could clarify their specific impact.
>
> > The multi-modal detectors can  .. ways: (1) detection of fake inputs within a multimodal signal ... (2) one-model-fits-all  ...  if there is any evidence of benefit for the first .. .
>
> To our knowledge, standard joint representation learning uses separate encoders per modality. While some methods encode multimodal signals directly, we did not explore such approaches. Our focus is on detecting deepfake content across separately processed modalities rather than within a combined multimodal signal. Thus, we have not explicitly tested the first scenario but acknowledge it as an interesting future direction

---

> > ### Comment · Reviewer_a9mT · 2025-03-03
> >
> > Dear authors, many thanks for your response. I am now reading the discussion with the other reviewers, and below are a few follow-up comments on the rebuttal:
> >
> > "Our discussion above highlights that ``well-trained single-mode classifiers'' are irrelevant to the setup we consider, since we do not have access to a train set per method (assuming that the reviewer meant to train a separate classifier per generation method). If the reviewer was referring to uni-modal networks, we did not explore them, some may exhibit properties similar to universal multi-modal models."
> >
> > In fact I was referring to uni-modal networks, therefore the underlying concern was not about the generalisation performance to different generators. I thought that actually the part the authors needed to justify is exactly the multi-modal aspect: if it were not to bring any benefits comparing to uni-modal ones, it's not clear why shouldn't this setup be simplified to uni-modal architectures. In the current version, it seems like there are two aspects in the architecture which are not clearly disentangled: the multi-modality and the middle-layer generalisation. I would expect the authors demonstrate the impact of each of them in isolation. Otherwise, the conclusions do not give an idea about the influence of each of these two factors.
> >
> > "We report confidence intervals for our main results (SVM9). Specifically, we report results along with confidence intervals computed over five different runs. We include this result in the revised version."
> > Many thanks!
> >
> > "We agree that a more structured approach would strengthen our claims. We propose the following methodology: We use an augmented-jpeg-compress variation of the validation dataset, while training on a small sample size, choosing best performing k. The augmentation is for better approximate different distribution from the train set. Results show that 7<=k<=10 is optimal. Further details are on appendix A."
> >
> > I would propose to go further than that and give a set of actionable items in the conclusion. What do the experiments suggest to use to improve the detection generalisation? How is it linked to the experiments? I would think of something like a discussion section of, e.g., Tobaben et al (2024) which gives a set of takeaway messages.
> >
> > Tobaben et al (2024) On the Efficacy of Differentially Private Few-shot Image Classification TMLR, 2024

---

### Review · Reviewer_Hkdx · 2025-02-09

**Summary Of Contributions:**

This paper studies the problem of Deepfake detection, i.e., distinguishing real media from generated content and identifying the source of the fake media. The authors explore the use of intermediate layer representations from a pre-trained multi-modal encoder model as discriminative features for classifying real and fake content. They implement a system combining feature extraction from pre-trained multi-modal encoder  with a linear classifier/SVM for detecting generated content across image and audio benchmark datasets. Experimental results demonstrate that the intermediate layer features show promising potential in distinguishing real from fake media on the testing data domains.

**Audience:**

No

**Broader Impact Concerns:**

I am not aware of any additional concerns specific to this work beyond the broader ethical and societal considerations that apply to all machine learning systems.

**Claims And Evidence:**

No

**Requested Changes:**

Notations:
- The dependence on the layer is not clearly reflected in Equation 2.
- The definition of the modality in Section 5 differs from the one in Section 3.
- The subscripts $i$ and $j$ are used multiple times with different meanings, which could lead to confusion. It would be helpful to standardize their usage for clarity.

The technical depth, particularly in the discussion of the results, seems somewhat shallow. It may be helpful to provide a stronger argument for why the choice of feature extractor layers is the key factor (rather than focusing on the configuration of the subsequent classifier such as model structure, SVM hyperparameters).

I am cautious about the claim that your approach achieves state-of-the-art based solely on the results of SVM_9 outperforming others. If the results were obtained by evaluating performance across all layers and then selecting layer 9 for presentation, this effectively introduces a form of hyperparameter tuning using test set information. It would be more convincing to include a dedicated section discussing how this selection process can be done in a principled manner—for example, using a small sample from one data subset and demonstrating its transferability across different settings.

**Strengths And Weaknesses:**

**Strengths**
- The exploration of the multi-modal Deepfake detection problem is both practically relevant and important.
- The experiments are relatively extensive, with the setup mostly clearly stated, contributing to a positive impression regarding reproducibility.

**Weaknesses**
- The paper seems to primarily re-implement an existing idea, focusing mainly on ablating/varying the layers of a feature extractor. This content might be more suitable as a section in an appendix rather than forming the core of a full-length journal paper.
- The insights provided are somewhat limited, and the overall contribution does not seem to advance the field significantly.
- There are some inconsistencies in the notation that could be improved for greater clarity and readability (see below)

---

> ### Author Response · Authors · 2025-02-25
>
> We appreciate Reviewer Hkdx careful reading of our manuscript and their constructive feedback. Our responses to main points issued are as follows:
>
> > The paper seems to primarily re-implement an existing idea, focusing mainly on ablating/varying the layers of a feature extractor. This content might be more ... appendix rather than forming the core of a full-length journal paper.
>
> Respectfully, we disagree with the reviewer. Many highly-impactful and mostly empirical methods are variants on existing tools and methodologies. To give a few examples: (1) mixup: Beyond Empirical Risk Minimization by Zhang et al. 2017, used linear interpolation to generate new samples, while similar techniques have been considered in the literature before that paper. (2) simCLR: A Simple Framework for Contrastive Learning of Visual Representations by Chen et al. 2020, performed contrastive learning using common tools. We believe that our paper is particularly important as we suggest an uncommon view with respect to the community--while the last layers of neural networks are typically assumed to be discriminative, we challenge this assumption in our work. Consequently, we obtain an effective classifier by using the middle layers whose results are on-par or better than SOTA approaches.
>
> > The insights provided are somewhat limited, and the overall contribution does not seem to advance the field significantly.
>
> We demonstrate that multi-modal models capture discriminative information distinguishing real from generated content across various modalities and generative models. While the use of intermediate layers in deep networks has been explored in prior work, to the best of our knowledge, no existing study has shown that such models naturally encode discriminative features for real vs fake content across multiple modalities and datasets. This observation contributes to a broader understanding of multi-modal representation learning for generative content detection. Moreover, we design and implemented a simple yet effective classifier, whose results are on-par or better in comparison to SOTA deepfake detection tools.
>
> > There are some inconsistencies in the notation that could be improved for greater clarity and readability
>
> We resolved the notation issues in the revised version.
>
> > The technical depth, particularly in the discussion of the results, seems somewhat shallow. It may be helpful to provide a stronger argument for why the choice of feature extractor layers is the key factor
>
> The main hypothesis in our work is that extracting better features leads to improved deepfake detection. Thus, the primary objective of our work is to establish a generalized approach for extracting generalizable features. As demonstrated in Section 4, the intermediate layers of the model capture highly relevant information for distinguishing real from generated content. To emphasize the significance of feature selection, we deliberately use a simple linear classifier rather than more complex models, ensuring that performance improvements stem from the choice of feature extractor layers rather than additional modeling complexity. In particular, we expect the results to improve even further under a stronger classifier (however, this change could also be applied to previous work).
>     Furthermore, our results show substantial performance gains when selecting the appropriate feature layers. For instance, in the UFD (CVPR 2023) setting, the last layer is chosen, whereas MLP\_0 is equivalent to UFD with only a change in the feature layer—resulting in an absolute accuracy improvement of 6–7\%. This highlights that the key factor driving performance is indeed the selection of the feature extraction layer in comparison to other network layers.
>
> > I am cautious about the claim that your approach achieves state-of-the-art based solely on the results of SVM9 outperforming others. ... for example, using a small sample from one data subset and demonstrating its transferability across different settings.
>
> The design of our approach and the choice of k=9  is motivated by related work. Specifically, prior findings by Gandelsman et al. (2023) demonstrate that the last four layers of CLIP primarily capture high-level semantic information. Moreover, several works (e.g., Zeiler and Fergus 2013) showed that the first layers of neural network learn elementary structures such as edges. Since such information is less beneficial for our task, we discard them. To maintain consistency and simplicity in our approach, we frame our selection symmetrically around the middle layer, leading to the  k=9 setting.
> We  agree that a more structured approach would strengthen our claims. We propose the following methodology: We use an augmented-jpeg-compress variation of the validation dataset, while training on a small samples. The augmentation is for better approximate different distribution from the train set. Results show that 7<=k<=10 is optimal. Further details are on appendix A.

---

### Review · Reviewer_FDxx · 2025-02-15

**Summary Of Contributions:**

This work tackles the problem of generalisability of generated media detectors across multiple data modalities and generative models. To this end, the authors propose a universal classifier for synthetic media detection through the use of pre trained multi-modal models. For this, they first do a thorough experimental analysis supporting the claim that features from middle layers of the model are more useful than start and end layers. Building on this observation, they develop a method leveraging features from k middle layers of a pre trained network and then use either an SVM or a single-layer MLP for the synthetic media detection. The experiments on various benchmarks showcase the effectiveness of the approach.

**Audience:**

Yes

**Claims And Evidence:**

Yes

**Requested Changes:**

Please see Weaknesses

**Strengths And Weaknesses:**

# Strengths

* I really liked the thorough analysis provided in Section 4. This provides enough experimental evidence for the key claim made in the paper, i.e. middle layers are more useful.
* The paper is clearly written, easy to follow and the idea is intuitive and simple. The idea has a clear motivation and all the claims are supported by the experimental evidence.
* The evaluation is also somewhat extensive, and the proposed method beats the state-of-the-art in most cases.

# Weaknesses
* Though the authors consider GANs, Diffusion and autoregressive methods, some evaluation of newer flow matching models would strengthen the paper even further, if possible.
* The paper lacks some further investigation on how to choose ‘k’, i.e. the number of middle layers to be chosen? I presume it depends on the total number of layers in the model but it might also change depending on the type of the model, e.g. GAN, Diffusion, etc. Ideally, a simple formula to calculate ‘k’ from number of layers would be useful. It would also be interesting to see if this kind of formula is generalisable.
 * The paper also lacks a deeper theoretical intuition of why middle layers are useful. Though it is experimentally evident, it would be interesting to see a deeper theoretical perspective on why it is the case?

---

> ### Author Response · Authors · 2025-02-25
>
> We thank Reviewer FDxx for careful reading of our manuscript and their constructive feedback. In what follows, we would like to address the specific comments put forth by Reviewer FDxx.
>
> > Though the authors consider GANs, Diffusion and autoregressive methods, some evaluation of newer flow matching models would strengthen the paper even further, if possible.
>
> We appreciate the  suggestion to include flow matching models in our evaluation. However, as flow matching is a relatively recent approach, there is currently no standardized benchmark dataset available for generated content from these models.  Generating such datasets and conducting a comprehensive evaluation across all methods would require significant computational resources. Nonetheless, we acknowledge the importance of evaluating newer generative techniques and consider this an interesting direction for future work.
>
> > The paper lacks some further investigation on how to choose ‘k’, i.e. the number of middle layers to be chosen? I presume it depends on the total number of layers in the model but it might also change depending on the type of the model, e.g. GAN, Diffusion, etc. Ideally, a simple formula to calculate ‘k’ from number of layers would be useful. It would also be interesting to see if this kind of formula is generalisable.
>
> We acknowledge the reviewer's concern regarding principled layer selection and agree that a more structured approach would strengthen our claims. To address this, we propose the following data-driven methodology: During training, we have access to data from a single generative model. However, since models typically perform well on their training distribution but generalize poorly to other generative models, selecting \( k \) based solely on validation performance on the training set would be misleading.
> To mitigate this issue, we propose using only a small subset (\(<200\) samples) of the training dataset for model training, while validating on the rest of the images which are jpeg-compressed, thus better approximating unseen generative distributions. Our results indicate that layers in the range $\( 7 \leq k \leq 10 \)$ consistently yield the best performance. We included detailed explanation of this method at the appendix of the revised version.
> Regarding the relationship between \( k \) and the total number of layers in the model, we hypothesize that the optimal \( k \) is influenced by both model depth and architecture type (e.g., GAN, Diffusion, etc.). While we do not yet provide a closed-form formula for determining \( k \), our preliminary findings suggest that selecting layers in the middle-to-late stages of the network, but before the final semantic layers, is a robust heuristic. We plan to further investigate this relationship and assess its generalizability in future work.
>
> > The paper also lacks a deeper theoretical intuition of why middle layers are useful. Though it is experimentally evident, it would be interesting to see a deeper theoretical perspective on why it is the case?
>
> We briefly discuss this in Section 2 under ``Layer-wise analysis of deep networks.'' Prior work by Gandelsman et al. (2023) has shown that the last four layers of CLIP are highly relevant for semantic tasks. Since deepfake detection does not primarily rely on high-level semantic understanding—and such information may even be detrimental to the task—this motivated our decision to discard the final layers.
>     Additionally, it is well-established (see e.g., Zeiler and Fergus 2013) that early layers in deep networks capture low-level features such as edges and textures, which are less informative for distinguishing real from generated content. Given this aggregated knowledge in the community, we were motivated to study the intermediate layers, which strike a balance between low-level feature extraction and high-level semantic abstraction.

---

### Decision · Action_Editor_jbps · 2025-03-17

**Recommendation:** Reject

**Comment:**

Post rebuttal, all three reviewers recommended rejection/leaning rejection due to unresolved issues (see details in Claims and Evidence). Given these substantial concerns, the paper does not currently meet TMLR’s acceptance threshold.

**Audience:**

The reviewers broadly agreed that the topic of synthetic content detection is relevant to TMLR's audience.

**Claims And Evidence:**

The submission provides a detailed experimental analysis to support its primary claim—that intermediate layers of pre-trained multi-modal models capture discriminative information effective for distinguishing real from synthetic content. However, post-rebuttal, all three reviewers remained unconvinced. Reviewers emphasized critical unresolved issues, such as a lack of principled justification for selecting intermediate layers (Hkdx), ambiguity around the precise definition and choice of 'middle layers' (FDxx), potential overfitting through hyperparameter selection using test data (Hkdx), and unclear advantages of multi-modal approaches compared to simpler uni-modal methods (a9mT). These significant concerns were not adequately addressed, undermining the claims and validity of the evidence presented.